# Salivary α-Amylase as a Metabolic Biomarker: Analytical Tools, Challenges, and Clinical Perspectives

**DOI:** 10.3390/ijms26157365

**Published:** 2025-07-30

**Authors:** Gita Erta, Gita Gersone, Antra Jurka, Peteris Tretjakovs

**Affiliations:** Department of Human Physiology and Biochemistry, Riga Stradins University, LV-1007 Riga, Latvia

**Keywords:** salivary α-amylase, *AMY1* gene, metabolic biomarker, glucose homeostasis, visceral adiposity, starch metabolism, enzymatic assay, biomarker standardization, personalized nutrition

## Abstract

Salivary α-amylase, primarily encoded by the AMY1 gene, initiates the enzymatic digestion of dietary starch in the oral cavity and has recently emerged as a potential biomarker in metabolic research. Variability in salivary amylase activity (SAA), driven largely by copy number variation of AMY1, has been associated with postprandial glycemic responses, insulin secretion dynamics, and susceptibility to obesity. This review critically examines current analytical approaches for quantifying SAA, including enzymatic assays, colorimetric techniques, immunoassays, and emerging biosensor technologies. The methodological limitations related to sample handling, intra-individual variability, assay standardization, and specificity are highlighted in the context of metabolic and clinical studies. Furthermore, the review explores the physiological relevance of SAA in energy homeostasis and its associations with visceral adiposity and insulin resistance. We discuss the potential integration of SAA measurements into obesity risk stratification and personalized dietary interventions, particularly in individuals with altered starch metabolism. Finally, the review identifies key research gaps and future directions necessary to validate SAA as a reliable metabolic biomarker in clinical practice. Understanding the diagnostic and prognostic value of salivary amylase may offer new insights into the prevention and management of obesity and related metabolic disorders.

## 1. Introduction

Obesity is a complex, multifactorial disease characterized by a chronic state of metabolic dysregulation [1,2,3]. It is strongly associated with insulin resistance, impaired glucose homeostasis, low-grade inflammation, and increased cardiometabolic risk [4,5]. The global rise in obesity prevalence underscores the urgent need for reliable, non-invasive biomarkers that can facilitate early detection of metabolic disturbances and guide personalized intervention strategies [6,7].

Recent studies have drawn attention to salivary α-amylase (SAA)—an enzyme predominantly produced by the salivary glands and encoded by the AMY1 gene—as a potential metabolic biomarker [8,9]. Traditionally recognized for its role in the initial digestion of dietary starch in the oral cavity, SAA has emerged as a functionally relevant modulator of postprandial glycemia and insulin dynamics [10,11,12]. Interindividual variability in SAA, largely driven by AMY1 gene copy number variation [13], has been associated with differences in glycemic response, visceral adiposity, and susceptibility to obesity and related metabolic disorders [14].

This review aims to summarize the physiological and metabolic roles of salivary amylase in the context of obesity and glucose regulation, critically evaluate current analytical approaches for measuring SAA, highlight key methodological challenges that hinder its clinical implementation, and explore the translational potential of SAA as a biomarker for metabolic risk stratification and personalized nutritional interventions.

By integrating recent advances in physiology, analytical biochemistry, and clinical research, this review seeks to clarify the role of SAA at the interface between oral digestion and systemic metabolic regulation in obesity.

The relevance of SAA as a metabolic biomarker has gained attention due to its ease of measurement from non-invasive saliva samples, making it an attractive candidate for clinical and research applications. However, substantial heterogeneity exists in the methodologies used to quantify SAA, and questions remain regarding the biological interpretation, reproducibility, and standardization of these measurements across populations and study designs.

## 2. Biochemical and Physiological Background

### 2.1. Structure and Function of SAA

Salivary α-amylase (SAA) is a calcium-dependent endo-enzyme belonging to the glycoside hydrolase family 13 (GH13) and is classified under EC 3.2.1.1 in the BRENDA enzyme database [15]. It catalyzes the hydrolysis of α-1,4-glycosidic linkages in polysaccharides such as starch, glycogen, and related oligosaccharides, generating maltose, maltotriose, and limit dextrins [16]. Although EC 3.2.1.1 includes both salivary and pancreatic isoforms, SAA is encoded by the AMY1 gene and is synthesized and secreted primarily by the acinar cells of the parotid and submandibular glands [17]. The enzyme comprises multiple domains, including a catalytic (β/α)-barrel domain (Domain A), a small Domain B contributing to substrate binding and calcium coordination, and Domain C, which is implicated in structural stabilization and substrate specificity [18].

In humans, two isoforms of α-amylase are encoded by the AMY1 (salivary) and AMY2 (pancreatic) gene clusters located on chromosome 1p21 [19]. The SAA, a glycoprotein with a molecular mass of approximately 56–62 kDa, is the predominant isoenzyme in the oral cavity that serves as the first enzymatic step in dietary carbohydrate digestion [20]. Beyond its digestive role, emerging evidence suggests that SAA may influence postprandial glycemia, insulin dynamics, and orosensory signaling pathways.

### 2.2. Physiological Regulation of SAA Secretion

SAA is synthesized and secreted predominantly by the serous acinar cells of the parotid glands, with minor contributions from the submandibular glands [21]. Its expression and secretion are under dual autonomic regulation, involving both the sympathetic and parasympathetic branches of the autonomic nervous system.

The sympathetic–adreno-medullary (SAM) axis modulates protein-rich secretion, particularly SAA, via norepinephrine release from postganglionic sympathetic fibers. Binding of norepinephrine to β-adrenergic receptors—predominantly β1 and β2 subtypes—on acinar cells triggers exocytosis of amylase-containing secretory granules through cyclic AMP (cAMP)-mediated intracellular signaling cascades [22].

In contrast, the parasympathetic nervous system regulates fluid volume and ionic composition of saliva, which indirectly facilitates SAA function and delivery. Parasympathetic innervation of the parotid glands arises from the glossopharyngeal nerve (cranial nerve IX), which relays via the otic ganglion and postganglionic fibers traveling through the auriculotemporal branch of the mandibular nerve (V3). The submandibular and sublingual glands are innervated by the facial nerve (cranial nerve VII) via the chorda tympani and the submandibular ganglion. Parasympathetic signals utilize acetylcholine acting on muscarinic M1 and M3 receptors located on acinar and ductal epithelial cells. While M3 receptors primarily mediate fluid secretion through intracellular calcium signaling, M1 receptors contribute to modulating glandular responsiveness and electrolyte transport.

Although parasympathetic stimulation does not directly initiate SAA exocytosis, it creates an optimal fluidic and ionic environment essential for SAA transport, dilution, and enzymatic action.

At the genomic level, AMY1, the gene encoding salivary amylase, exhibits pronounced interindividual copy number variation (CNV), which has been strongly associated with both basal and stimulated SAA output [23]. This genetic variation interacts with environmental modulators—including dietary starch intake, autonomic tone, and psychosocial stress—highlighting the importance of integrating both genomic and physiological variables when interpreting SAA activity in clinical or experimental contexts.

### 2.3. Factors Affecting SAA

SAA activity exhibits pronounced temporal and situational variability. One of the primary modulators is the circadian rhythm, wherein SAA concentrations demonstrate a diurnal pattern: levels are typically low upon awakening, followed by a sharp increase (the “morning surge”) and gradual decline throughout the day, independent of salivary flow rate [24]. This profile must be accounted for when standardizing sampling protocols.

Acute and chronic psychological stress are robust inducers of SAA secretion. Activation of the sympathetic nervous system results in a rapid elevation of SAA levels, rendering it a valuable non-invasive surrogate marker of adrenergic reactivity. However, individual differences in stress responsiveness and habituation can affect its reliability as a biomarker [22].

Moreover, pathophysiological conditions influence SAA expression and activity. For instance, individuals with metabolic syndrome, diabetes mellitus, or obesity often display altered SAA profiles, potentially reflecting neuroendocrine dysregulation. Similarly, oral health status, including salivary gland inflammation, xerostomia, or periodontal disease, can affect both qualitative and quantitative aspects of SAA secretion [25].

Nutritional status, physical exercise, smoking, medication use (e.g., β-blockers, corticosteroids), and even hydration state further modulate SAA [26]. Therefore, interpretation of SAA activity requires consideration of a multifactorial framework, encompassing genetic, physiological, behavioral, and environmental variables.

### 2.4. Metabolic Relevance of Salivary Amylase in Obesity

#### 2.4.1. Influence on Postprandial Glycemia and Insulin Dynamics

Salivary amylase plays a critical role in modulating the rate and extent of carbohydrate digestion in the cephalic phase, thereby shaping postprandial glycemic responses [27]. Individuals with higher salivary amylase activity (SAA), often reflecting increased AMY1 gene copy number, exhibit more efficient hydrolysis of dietary starch into oligosaccharides and disaccharides such as maltose [28]. This enhanced starch pre-digestion has been associated with altered glycemic excursions and insulin secretion patterns.

Paradoxically, elevated SAA has been linked both to improved glucose tolerance in some populations and to exaggerated early-phase insulin responses in others, suggesting a dual, context-dependent metabolic impact [29]. Recent studies indicate that SAA may influence not only the magnitude but also the kinetics of insulin secretion, particularly affecting the first-phase insulin response [24]. This altered insulin dynamic may contribute to variations in insulin sensitivity and β-cell stress, potentially predisposing certain individuals to compensatory hyperinsulinemia [30] (Figure 1).

#### 2.4.2. Association with Visceral Adiposity and Metabolic Phenotypes

A growing body of evidence supports a link between SAA and adipose tissue distribution, particularly visceral adiposity [31]. Low SAA levels have been associated with higher visceral fat accumulation, independent of total body mass index (BMI), indicating a specific association with metabolically adverse fat depots. Visceral adipose tissue is known to be more lipolytically active and pro-inflammatory, contributing to insulin resistance via adipokine secretion and ectopic lipid deposition [32].

Mechanistically, it is hypothesized that altered starch processing capacity, as indexed by SAA, may influence substrate availability and hormonal signaling pathways involved in energy storage and appetite regulation [4]. Individuals with low SAA may experience delayed starch digestion and prolonged glycemic load, potentially driving increased insulin secretion and fat accumulation over time.

#### 2.4.3. Role in the Pathophysiology of Insulin Resistance and Glucose Intolerance

SAA may be implicated in the pathogenesis of insulin resistance and glucose intolerance through its modulation of postprandial insulin dynamics and nutrient sensing pathways. Chronic hyperinsulinemia, which may result from exaggerated insulin responses to rapidly absorbed starch in high-SAA individuals or from inefficient digestion and prolonged glucose absorption in low-SAA individuals, can downregulate insulin receptor sensitivity and disrupt metabolic homeostasis [33].

At the molecular level, salivary amylase-mediated differences in starch digestion may engage several intracellular signaling cascades, including the mechanistic target of rapamycin (mTOR) pathway, which integrates nutrient availability with cellular growth, metabolism, and insulin action [34]. Hyperactivation of mTOR complex 1 (mTORC1) in response to repeated postprandial hyperinsulinemia may contribute to negative feedback inhibition of insulin receptor substrate (IRS) proteins, particularly IRS-1, impairing downstream insulin signaling via the PI3K-Akt pathway [35]. This molecular mechanism has been implicated in hepatic and skeletal muscle insulin resistance.

Additionally, frequent postprandial activation of mTOR may influence β-cell mass and function. While transient mTOR activation supports β-cell compensation during metabolic stress, chronic stimulation can lead to β-cell dysfunction, oxidative stress, and apoptosis, thereby exacerbating glucose intolerance. The interplay between SAA, insulin secretion, and mTOR signaling thus represents a potential mechanistic axis linking oral starch processing with systemic metabolic dysregulation.

Moreover, SAA-driven variations in glucose and insulin kinetics may impact other nutrient-sensing systems, such as the AMP-activated protein kinase (AMPK) pathway and gut-derived incretins (e.g., GLP-1), further modulating energy balance and metabolic health [36].

## 3. Methodological Considerations in the Assessment of Salivary Amylase Activity

Salivary amylase activity (SAA) assessment is characterized by considerable methodological heterogeneity, influenced by the analytical platform, reaction conditions, and sample collection strategy (Table 1).

### 3.1. Colorimetric Assays

#### 3.1.1. 2-Chloro-4-nitrophenyl-α-d-maltotrioside Substrate

This method utilizes the cleavage of 2-chloro-4-nitrophenyl-α-d-maltotrioside by salivary α-amylase, releasing a colored byproduct. The assay is optimized for concentrations between 20 and 500 μg/mL with a limit of detection (LOD) of 8 μg/mL.

A paper-based strip assay is proposed for rapid and specific detection, with minimal interference from saliva components. RGB analysis offers quantitative detection with an LOD of 11 μg/mL [37].

#### 3.1.2. 3,5 Dinitrosalicylic Acid Assay

This method quantifies the reducing sugar released from soluble starch via α-amylase hydrolysis, producing a brick-red product measured at 525 nm. A hand-held device using this method shows excellent correlation with commercial spectrophotometers and measures α-amylase activity in the range of 0.1–1.0 U/mL [38].

#### 3.1.3. Cu/Au Nanoclusters

A novel technique using starch-stabilized Cu/Au nanoclusters that exhibit peroxidase-like activity. The presence of α-amylase leads to starch digestion, causing nanocluster aggregation and decreased peroxidase activity, detectable by a color change. This method has a detection limit of 0.04 U/mL and a linear range of 0.1–10 U/mL [38].

#### 3.1.4. iPhone Imaging with Dinitrosalicylic Acid Assay Method

This approach employs iPhone imaging and Adobe Photoshop for colorimetric analysis. It provides comparable sensitivity and linearity to spectrophotometric methods, with better inter-day precision [39].

### 3.2. Spectrophotometric Assays

#### 3.2.1. Phadebas Test

A widely used spectrophotometric method for detecting α-amylase activity. It involves a colorimetric endpoint assay and is benchmarked against newer methods for accuracy and reliability [40].

#### 3.2.2. Automated Kinetic Spectrophotometric Method

This method uses standard reagents for pancreatic amylase activity and shows excellent correlation with manual colorimetric assays. It allows for standardized, center-independent analyses [41] (Table 2).

### 3.3. Fluorometric Methods

Provide high sensitivity and lower detection limits compared to colorimetric assays; Often use 4-methylumbelliferyl-α-D-glucopyranoside as substrate; Require fluorescence detection instruments, limiting point-of-care use.

#### 3.3.1. Nano CdS Doped Sol-Gel Matrix Method

This luminescence-based method utilizes a nano-CdS doped sol–gel matrix, where the emission intensity at 634 nm is quenched by maltose—the product of α-amylase-mediated starch hydrolysis. The luminescence quenching is proportional to maltose concentration, allowing indirect quantification of α-amylase activity. The method demonstrates a wide linear calibration range (4.8 × 10^−10^ to 1.2 × 10^−5^ mol·L^−1^) and an exceptionally low detection limit of 5.7 × 10^−11^ mol·L^−1^ [42]. This approach offers notable advantages, including high sensitivity and precision, making it suitable for trace-level detection. It has been successfully applied in the analysis of α-amylase activity in human saliva samples, highlighting its potential for clinical and biochemical diagnostics.

#### 3.3.2. Starch–Iodine–Sodium Fluorescein Complex (SIF) Method

This method is based on the formation of a ternary starch–iodine–sodium fluorescein (SIF) complex, which exhibits low fluorescence under native conditions. Upon enzymatic hydrolysis of starch by α-amylase, the complex is disrupted, resulting in a measurable increase in fluorescence emission intensity. The method demonstrates a linear calibration range for α-amylase activity between 0.18 and 9.00 U/L [43]. It offers several advantages, including low cost, operational simplicity, and adequate sensitivity. This assay has been effectively applied to the detection of α-amylase activity in a range of biological matrices.

#### 3.3.3. Aggregation-Induced Emission Luminogens (AIEgens) Method

AIEgens with D-π-A structures are used where the fluorescence emission is significantly enhanced upon interaction with α-amylase. The limit of detection (LOD) for this method is 0.1864 U/L, and it offers high specificity and selectivity [44].

#### 3.3.4. Starch-Coated Fullerene C60 Complex Method

This method utilizes the quenching of Triphenylphosphine oxide (TPPOH) fluorescence by starch-coated fullerene C60. The analytical response shows a linear fluorescent response in α-amylase concentrations ranging from 0.001 to 0.1 Units/mL, with an LOD of 0.001 Units/mL [45]. This method provides high sensitivity and is applicable to clinical samples, making it suitable for caries detection and risk assessment.

#### 3.3.5. Fluorescence Spectroscopy for Forensic Analysis

This method detects saliva stains on inanimate objects by comparing the fluorescence emission spectra of dried saliva samples to undiluted liquid saliva. The emission peak around 350 nm is used to identify the presence of amylase [46].

This method is simple and effective for forensic identification and is used for screening and selecting samples for subsequent DNA analysis (Table 3).

### 3.4. Emerging Technologies

#### 3.4.1. Biosensors for Salivary Amylase Detection

##### Piezoresistive Microcantilever Biosensor

This biosensor detects salivary amylase activity by measuring the deflection of a microcantilever beam upon interaction with the enzyme. The deflection causes a change in resistance, which is measured using a Wheatstone Bridge circuit, converting the biochemical signal into a measurable voltage signal [47].

##### Flat-Chip Microanalytical Enzyme Sensor

Designed for wearable systems, this sensor incorporates enzymatic membranes on a small flow cell. It can measure amylase activity in a sample volume of 50 microliters with high sensitivity, making it suitable for continuous monitoring [48,49].

##### Amperometric Biosensor

This sensor uses salivary antibodies or antigens self-assembled onto an Au-electrode. The interaction between the immobilized antibody and salivary amylase is monitored via an electroactive indicator, providing analytical information based on current changes [49].

##### Smartphone-Based Potentiometric Biosensor

This system includes a smartphone app, a potentiometric reader, and a sensing chip. The saliva sample reacts with preloaded reagents on the chip, and the resulting potential is measured and converted into amylase concentration by the app [50].

##### Tri-Enzymatic Biosensor

Utilizing a screen-printed electrode modified with Prussian Blue, this biosensor co-immobilizes α-glucosidase, glucose oxidase, and mutarotase. It measures the maltose generated by the hydrolysis of maltopentose in the presence of salivary amylase [51].

#### 3.4.2. Microfluidic Devices for Salivary Amylase Detection

##### Paper-Based Microfluidic Chip

This device isolates α-amylase from saliva using a starch-coated paper-based chip. The concentration of α-amylase is determined by comparing the enzyme concentration in different sections of the chip [52].

#### 3.4.3. Lab-on-a-Chip (LOC) Devices

LOC devices integrate biosensors and microfluidics to analyze small sample quantities efficiently. These devices are particularly useful for non-invasive saliva analysis, offering high throughput, portability, and disposability [53,54].

##### Dual Microfluidic Paper-Based Analytical Devices (Dual-μPADs)

These devices combine colorimetric and electrochemical modules to detect salivary amylase along with other biomarkers. They are fabricated using a simple “do-it-yourself” protocol, making them versatile and user-friendly for point-of-care diagnostics [55] (Table 4).

Innovative platforms have emerged to enable portable, rapid, and minimally invasive testing.

Electrochemical biosensors show promise for real-time detection, with enzyme immobilization techniques enhancing stability.

Microfluidic paper-based analytical devices (μPADs) combine colorimetric readouts with saliva capillary flow, offering low-cost diagnostic potential.

Integration with smartphone-based readers was reported in recent studies to improve field applicability.

Salivary amylase activity presents both physiological advantages and inherent limitations as a biomarker, particularly in the context of stress response and metabolic regulation (Table 5).

## 4. Key Innovations and Developments

### 4.1. Miniaturization and Sensitivity

A flat-chip microanalytical enzyme sensor has been developed, incorporating a flow cell as small as a C battery. This sensor uses enzymatic membranes containing maltose phosphorylase and glucose oxidase immobilized on a planar surface, allowing for the detection of amylase activity in the range of 0–190 kU/L with a sample volume of 50 μL [49,56].

Another approach involves a smartphone-based potentiometric biosensor that uses a sensing chip with preloaded reagents. This system can quantitatively analyze salivary α-amylase within 5 min, correlating well with psychological states [57].

### 4.2. Wearable and Portable Systems

A completely automated hand-held monitor for salivary α-amylase activity has been developed, utilizing a dry-chemistry system with a disposable test strip. This device can measure amylase activity with high accuracy using only 30 μL of saliva, making it suitable for continuous monitoring and psychological research [52].

### 4.3. Microfluidic and Paper-Based Chips

A microfluidic starch-coated paper-based chip has been designed to isolate α-amylase from human saliva. This chip effectively concentrates α-amylase in specific sections, aiding in its detection using techniques like Western blotting and ELISA [58].

Another innovative design includes a polymer LOC with dried on-chip immunoassay reagents for detecting unbound cortisol in saliva, demonstrating the versatility of LOC systems for various biomarkers [59].

### 4.4. Integration with Modern Technologies

LOC systems often integrate microfluidic elements for fluid mixing, manipulation, and control, enabling the performance of conventional laboratory procedures on a miniaturized chip. These systems can be developed on paper or polymeric platforms using various fabrication techniques [60]

LOC systems for salivary amylase detection are particularly useful for non-invasive monitoring of the sympathetic nervous system and stress-related conditions [61,62,63]

They also hold promise for clinical diagnostics, forensic applications, and personalized health care by providing rapid, accurate, and cost-effective analysis of salivary biomarkers [43,64,65].

In summary, lab-on-a-chip systems for detecting salivary amylase activity represent a significant advancement in non-invasive diagnostics, offering high sensitivity, portability, and the potential for continuous monitoring in various health-related applications.

## 5. Discussion

This comprehensive review provides an in-depth analysis of analytical methods used for the detection of salivary amylase activity (SAA), encompassing both conventional enzymatic assays and novel biosensor technologies. The findings reveal considerable methodological heterogeneity across studies, which impacts analytical performance, data comparability, and the clinical applicability of SAA as a biomarker.

### 5.1. Analytical Performance Comparison

Analytical performance varied substantially across reported methodologies. Detection limits ranged from 0.01 to 50 U/mL, indicating significant differences in assay sensitivity. Intra-assay coefficients of variation ranged from 2% to 20%, reflecting inconsistencies in precision. Furthermore, calibration against recognized international reference standards, such as those of the International Federation of Clinical Chemistry (IFCC), was rare, limiting methodological comparability and the development of diagnostic cut-offs.

Traditional spectrophotometric assays (e.g., starch–iodine, dinitrosalicylic acid [DNS] methods) remain widely used due to their simplicity and low cost. However, these methods often exhibit suboptimal specificity and limited dynamic range in complex matrices like saliva [49,66,67]. Enzymatic assays using chromogenic or fluorogenic substrates demonstrate enhanced sensitivity and kinetic resolution but are hindered by a lack of standardized reagents and protocols [68].

Innovative biosensors and microfluidic platforms offer a promising future for real-time, point-of-care detection of SAA. These systems show potential for miniaturization, multiplexing, and integration with digital health tools. However, their clinical translation remains limited due to device fabrication complexity, sample matrix interference (e.g., saliva viscosity and contaminants) and a paucity of large-scale validation studies [69].

### 5.2. Pre-Analytical Considerations

Pre-analytical variability emerged as a major factor influencing SAA measurement reliability. Saliva collection methods differed considerably between studies, with both stimulated (e.g., chewing paraffin) and unstimulated (e.g., passive drool, swab) techniques employed. These variations directly affect enzyme concentration, volume, and sample composition [70,71,72].

Storage conditions were inconsistently reported—some studies applied immediate freezing or used preservatives, while others lacked detailed documentation. Additionally, physiological variables such as diurnal fluctuation, food intake, and acute stress were often uncontrolled, despite their known influence on SAA dynamics [73,74]. This pre-analytical inconsistency weakens inter-study comparisons and diminishes the biomarker’s clinical and research validity.

### 5.3. Clinical and Research Applications

Beyond its classical digestive role, SAA has emerged as a promising biomarker across multiple physiological domains. In psychoneuroendocrinology, it serves as a non-invasive indicator of sympathetic nervous system activation in response to acute and chronic stressors [75,76]. It has also been explored as a surrogate marker for autonomic dysfunction in cardiovascular and psychiatric disorders [77,78].

Emerging evidence links SAA with metabolic outcomes, including obesity and insulin resistance, suggesting its potential as a biomarker of metabolic flexibility and resilience. Additionally, its role in oral health diagnostics is under investigation, particularly in relation to the oral microbiome and periodontal inflammation [79]. However, the clinical implementation of SAA remains limited by the absence of standardized protocols, validated thresholds, and reference ranges across populations.

### 5.4. Integration with Multi-Biomarker Panels

The integration of SAA into multiplex biomarker platforms offers a powerful strategy to enhance the diagnostic specificity of saliva-based assessments for both psychophysiological stress and metabolic disorders. While SAA serves as a dynamic biomarker of sympathetic–adrenal–medullary (SAM) axis activation, its diagnostic accuracy can be augmented by combining it with complementary salivary biomarkers representing distinct physiological axes, including hormonal, immunological, and metabolic pathways.

Notably, salivary cytokines—such as interleukin-6 (IL-6), tumor necrosis factor-alpha (TNF-α), and interleukin-1β (IL-1β)—can be co-analyzed with SAA to assess inflammatory states associated with chronic stress, metabolic syndrome, or low-grade systemic inflammation. These cytokines, when detected in saliva, offer insight into mucosal immune activation and neuroimmune crosstalk, contributing to a broader understanding of stress-related pathophysiology.

Moreover, the inclusion of salivary steroid hormones such as cortisol, dehydroepiandrosterone (DHEA), and testosterone further refines the interpretation of stress responses by capturing hypothalamic–pituitary–adrenal (HPA) axis activity. For example, the simultaneous measurement of SAA (SAM axis) and cortisol (HPA axis) provides a dual-axis framework to distinguish between acute and chronic stress, as well as dysregulations in stress reactivity.

Additionally, the incorporation of metabolic hormones and analytes detectable in saliva—such as insulin, leptin, ghrelin, and adiponectin—may facilitate early, non-invasive detection of metabolic impairments. This integrative approach holds promise for profiling metabolic flexibility, energy balance, and appetite regulation in conditions such as obesity, type 2 diabetes, or polycystic ovary syndrome (PCOS).

Emerging lab-on-chip technologies and electrochemical biosensors increasingly allow for the concurrent quantification of such panels from minimal sample volumes. The development of salivary multi-biomarker platforms may thus revolutionize point-of-care diagnostics by providing real-time, holistic insight into systemic physiological status.

### 5.5. Future Directions

The continuous evolution of analytical technologies for salivary α-amylase (SAA) detection expands the potential for clinical diagnostics, psychophysiological research, and personalized medicine. Despite significant advances in colorimetric, spectrophotometric, fluorometric, and biosensor-based methodologies, several challenges remain to be addressed to optimize the sensitivity, specificity, and field applicability of these techniques.

#### 5.5.1. Integration and Miniaturization

Emerging lab-on-a-chip (LOC) systems and microfluidic paper-based analytical devices (µPADs) have demonstrated substantial promise for point-of-care (POC) diagnostics. Future research should focus on enhancing the integration of multistep assays (e.g., enzymatic reactions, signal amplification, and detection) within single miniaturized platforms. This would allow for fully automated, real-time monitoring of salivary biomarkers with minimal sample volume requirements and without the need for sophisticated laboratory infrastructure.

#### 5.5.2. Multiplexed Detection and Biomarker Panels

Given the complex physiology of stress and metabolic regulation, future efforts should aim at developing platforms capable of simultaneous detection of SAA alongside complementary biomarkers, such as cortisol, glucose, and inflammatory cytokines. Dual-mode biosensors (e.g., combining colorimetric and electrochemical outputs) and multi-analyte LOC systems offer a promising strategy to improve diagnostic accuracy and broaden clinical utility.

#### 5.5.3. Wearable Technologies and Remote Monitoring

The development of wearable biosensors that enable continuous, non-invasive monitoring of SAA in real-life environments is a critical frontier. Flexible, skin-mounted platforms or intraoral sensors integrated with wireless data transmission and smartphone interfaces could significantly enhance our ability to monitor stress, fatigue, and metabolic responses dynamically in ambulatory settings.

#### 5.5.4. AI-Driven Signal Analysis and Decision Support

The integration of artificial intelligence (AI) and machine learning algorithms into biosensor platforms can transform raw data into actionable insights. Future systems could incorporate real-time pattern recognition to predict stress episodes or detect deviations in metabolic profiles, thus enabling preventive interventions and personalized health recommendations.

#### 5.5.5. Standardization and Clinical Translation

To bridge the gap between laboratory innovation and clinical implementation, future studies should emphasize assay standardization, inter-laboratory validation, and regulatory approval. Harmonization of detection units, sample collection protocols, and calibration procedures are essential for the broad adoption of SAA-based diagnostics in clinical practice.

#### 5.5.6. Expanding Diagnostic Applications

Although SAA is widely recognized as a surrogate marker of sympathetic nervous system activity, future research should explore its potential role in broader diagnostic contexts, including metabolic syndrome, oral diseases (e.g., caries risk), gastrointestinal disorders, and early detection of neuropsychiatric conditions. Novel assay designs and longitudinal studies are required to fully realize the biomarker potential of SAA in systemic health monitoring.

In conclusion, the future of salivary α-amylase detection lies in interdisciplinary innovation—combining biochemical engineering, materials science, data analytics, and clinical research—to develop robust, accessible, and informative diagnostic platforms for health care and beyond.

### 5.6. Limitations

This review is limited by the heterogeneity of included studies, which precluded meta-analytic synthesis. Moreover, potential publication bias may have favored the reporting of studies with positive or novel findings while underreporting those with null results or technical failures.

## 6. Concluding Recommendations and Best-Fit Methodologies

Given the increasing interest in salivary α-amylase (SAA) as a non-invasive biomarker at the intersection of metabolic, autonomic, and stress-related physiology, careful alignment of methodological approaches with specific research and clinical objectives is essential. Although a wide array of analytical platforms exists for measuring SAA, their applicability, reproducibility, and translational relevance vary significantly depending on the context of use.

To support researchers and clinicians in selecting appropriate methods, we summarize the current state of evidence regarding best-fit platforms for three key domains of application: (1) wearable stress monitoring, (2) metabolic phenotyping in obesity and insulin resistance, and (3) forensic or retrospective physiological assessments.

### 6.1. Wearable Stress Monitoring

For dynamic assessment of autonomic reactivity, particularly in ambulatory or ecologically valid conditions, real-time biosensing technologies based on enzymatic electrochemical detection of SAA are considered most suitable. These platforms offer high temporal resolution and direct assessment of enzymatic activity, correlating well with sympathetic nervous system activation. In contrast, immunoassays (e.g., ELISA) that quantify total SAA protein lack temporal resolution and functional specificity and are therefore not recommended for stress reactivity studies where rapid fluctuations are of interest [80].

### 6.2. Metabolic Phenotyping in Obesity

SAA activity shows emerging promise as a metabolic biomarker related to postprandial glucose regulation, visceral adiposity, and insulin sensitivity. In this context, kinetic enzymatic assays using colorimetric or fluorometric readouts remain the gold standard, particularly when combined with pre-analytical standardization (e.g., fasting state, collection time, flow rate normalization). While AMY1 gene copy number has been proposed as a genetic proxy for SAA levels, its application is limited by poor functional correlation, interindividual variability, and platform-dependent detection errors [81]. Thus, its use is acceptable only with caution and should ideally be supplemented by concurrent enzymatic activity measurement.

### 6.3. Forensic and Retrospective Assessment

In contexts where only archived or post hoc biological samples are available, the use of total protein quantification (e.g., Western blot, ELISA) may be pragmatically justified. However, these methods do not reflect enzymatic function and are susceptible to degradation and post-collection variability. Therefore, their use should be interpreted cautiously and ideally complemented by additional salivary biomarkers—such as cortisol (a marker of HPA axis activation), selected pro-inflammatory cytokines (e.g., IL-6, TNF-α) that reflect low-grade systemic inflammation, or adipokines like leptin and resistin, which are increasingly detectable in saliva and provide insight into the inflammatory–metabolic interface characteristic of obesity-related pathophysiology (Table 6).

## 7. Conclusions

Salivary amylase activity has emerged as a promising non-invasive biomarker with applications in stress physiology, metabolic health, and oral diagnostics. While traditional colorimetric and spectrophotometric methods remain widely used, recent advances in biosensors, microfluidics, and fluorometric assays offer improved sensitivity, portability, and diagnostic potential. However, significant variability in sample collection, assay protocols, and reporting standards continues to limit comparability across studies and hinders clinical translation. Future progress will depend on the development of standardized methodologies, validation in large cohorts, and integration of emerging technologies into clinical workflows. By addressing these challenges, SAA measurement could become a valuable component of precision health monitoring and personalized diagnostics.

## Figures and Tables

**Figure 1 ijms-26-07365-f001:**
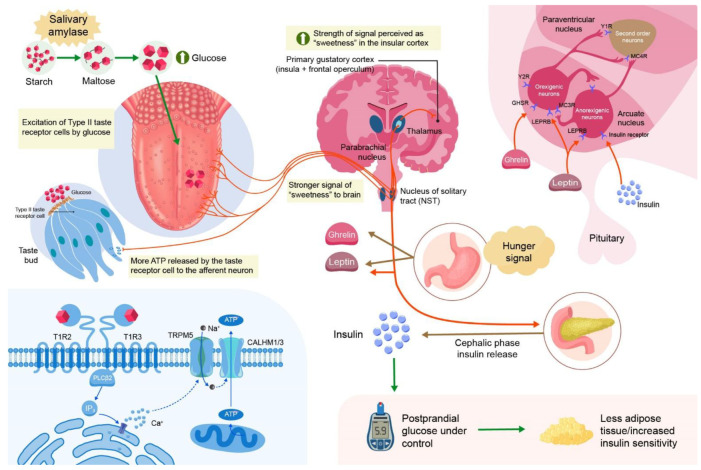
SAA influence on postprandial glycemia and insulin dynamics.

**Table 1 ijms-26-07365-t001:** Overview of methodological heterogeneity in salivary amylase activity assessment.

Methodological Category	Analytical Technique	Analytical Range/Precision	Remarks
Dry Chemistry Platforms	Test strips with optical analyzers	0–200 kU/L	Fixed reaction time enhances reproducibility.
Electrochemical Biosensors	Enzyme-based electrodes; flow-injection systems	0–30 kU/L	Demonstrates high specificity and analytical sensitivity.
Portable Point-of-Care Devices	Saliva strip application or pipetting	Variable, method-dependent	Pipetting yields more consistent results than direct strip contact.
Spectrophotometric Assays	Colorimetric reaction with CNPG3 substrate	High precision, broad dynamic range	Allows differentiation of amylase isoforms (proteoforms).
Competitive or Product Inhibition Assays	Modified dry-chemistry platforms	Extended analytical range	Offers cost-effective high-throughput potential.
Sample Collection Devices	Salivette systems	Good intra-assay precision	Saliva recovery and analyte stability may vary with matrix.
Forensic Detection Methods	RSID™-Saliva immunochromatographic assay	High sensitivity, qualitative	Effective in detecting amylase in degraded or minute samples.

**Table 2 ijms-26-07365-t002:** Comparison of Colorimetric and Spectrophotometric Methods.

Method	Type	Detection Limit	Range	Advantages
2-Chloro-4-nitrophenyl-α-d-maltotrioside	Colorimetric	8 μg/mL	20–500 μg/mL	Rapid, specific, paper-based strip
Dinitrosalicylic Acid Assay (Hand-held)	Colorimetric	-	0.1–1.0 U/mL	Portable, cost-effective
Cu/Au Nanoclusters	Colorimetric	0.04 U/mL	0.1–10 U/mL	High selectivity, affordable
iPhone Imaging (Dinitrosalicylic Acid Assay)	Colorimetric	-	-	Field and lab use, high precision
Phadebas Test	Spectrophotometric	-	-	Standard method, reliable
Automated Kinetic Spectrophotometric	Spectrophotometric	-	-	Standardized, high correlation

**Table 3 ijms-26-07365-t003:** Comparison of Fluorometric Methods.

Method	Principle	Sensitivity	Applications	Advantages
Nano CdS Doped Sol-Gel Matrix	Quenching of luminescence by maltose	5.7 × 10^−11^ mol L^−1^	Human saliva samples	High sensitivity and precision
SIF Complex	Decomposition of SIF complex increases fluorescence	0.18–9.00 U/L	Biological samples	Inexpensive, easy to use
AIEgens	Enhanced fluorescence upon interaction with α-amylase	0.1864 U/L	Sensory experience assessment	Rapid, high reliability
Starch-Coated Fullerene C60	Quenching of TPPOH fluorescence	0.001 Units/mL	Caries detection	High sensitivity
Fluorescence Spectroscopy	Emission spectra comparison	N/A	Forensic analysis	Simple, effective

**Table 4 ijms-26-07365-t004:** Comparison of Emerging Technologies.

Method	Principle	Sample Volume	Sensitivity	Application
Piezoresistive Microcantilever	Resistance change	Not specified	High	Stress detection
Flat-Chip Sensor	Enzymatic reaction	50 µL	High	Wearable systems
Amperometric Biosensor	Current change	Not specified	1.57 pg/mL	Real-time monitoring
Smartphone-Based Potentiometric	Potential measurement	Not specified	High	Point-of-care testing
Tri-Enzymatic Biosensor	Enzymatic reaction	Not specified	5 U/mL	Simple assays
Paper-Based Microfluidic Chip	Enzyme concentration	Not specified	High	Non-invasive diagnostics
Dual-μPADs	Colorimetric and Electrochemical	Not specified	High	Periodontal disease diagnosis

**Table 5 ijms-26-07365-t005:** Physiological Advantages and Limitations of Salivary Amylase Activity as a Biomarker.

Advantages	Limitations
Noninvasive, rapid sampling method—suitable for repeated measures and ambulatory settings.	High sensitivity to confounding variables—affected by circadian rhythm, diet, hydration, and oral hygiene.
Potential biomarker of autonomic nervous system (ANS) activation—particularly sympathetic–adrenal–medullary (SAM) axis responsiveness.	Insufficient large-scale clinical validation—limited normative data across populations and disease states.
Responsive to acute psychological and physiological stress—may reflect real-time stress-related physiological dynamics.	Marked inter-individual variability—influenced by genetic (AMY1 gene copy number), metabolic, and environmental factors.
Advances in enzyme detection technologies—allow for accurate, low-volume, and point-of-care measurements.	Comparative biomarker uncertainty—less standardized and validated compared to established stress or metabolic biomarkers.

**Table 6 ijms-26-07365-t006:** Summary of Methodological Suitability.

Application	Methodological Approach	Classification	Rationale
Wearable Stress Monitoring	Real-time enzymatic biosensors (electrochemical)	Recommended	High temporal resolution, functional specificity to sympathetic activity, real-time feedback.
	Point-sample enzymatic assay (e.g., spectrophotometric)	Conditionally Appropriate	Useful in lab settings; limited ecological validity and temporal resolution.
	Immunoassays (e.g., ELISA for total SAA protein)	Not Recommended	Low functional specificity; poor correlation with stress reactivity and enzyme activity.
Metabolic Phenotyping (e.g., Obesity, Insulin Sensitivity)	Enzymatic activity assay (e.g., kinetic, chromogenic)	Recommended	Reproducible in fasting/postprandial states; associated with glucose-insulin dynamics.
	AMY1 gene copy number estimation (e.g., qPCR, ddPCR)	Conditionally Appropriate	Moderate heritability; limited functional correlation due to CNV complexity.
	Total protein concentration (e.g., western blot)	Not Recommended	Does not reflect enzymatic function; poor metabolic specificity.
Forensic/Retrospective Analysis	Total SAA protein (e.g., western blot, ELISA)	Conditionally Appropriate	Useful when only preserved or archived saliva is available; low functional precision.
	Enzymatic activity (archived or stored samples)	Recommended	Preferred when integrity is preserved; enzyme stability is time- and storage-dependent.
	AMY1 copy number (retrospective genetic profiling)	Conditionally Appropriate	Stable DNA allows retrospective genotyping; functional extrapolation is uncertain.

## Data Availability

Not applicable.

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
