# Peer review of "Salivary α-Amylase as a Metabolic Biomarker: Analytical Tools, Challenges, and Clinical Perspectives"

_ijms, 2025, doi:10.3390/ijms26157365_

Round 1

Reviewer 1 Report

Comments and Suggestions for Authors

The paper entitled " Salivary α-amylase as a Metabolic Biomarker in Obesity: Analytical Tools, Challenges, and Clinical Perspectives" provides a comprehensive review of salivary α-amylase (SAA) as a potential metabolic biomarker in the context of obesity. The authors thoroughly discuss the biochemical basis, methodological approach to measuring SAA, and perspectives for clinical and research applications.

The manuscript gives the technical overview of analytical systems concerning the detection of salivary α-amylase and forgoes the functional integration of some analytical methods concerning their relevance to particular clinical or research functions.

This review would be far more useful if it included a concluding recommendation section with “best fit” methods to document different diagnostic or investigational uses such as wearable stress monitors, phenotyping obesity, and forensic evaluation. For example, which methods should be considered ill-advised to use because of low reproducibility, specificity, or failure to translate clinically?

This kind of integration would save researchers and practitioners alike from trying to validate or analytically test unreliable platforms.

Also, please do a technical check of the manuscript. For example, literature reference number 17. and 41. are missing. Also, please pay attention to the numbering of the subheadings, especially take a close look at Chapter 3.

Author Response

We thank the reviewer for the thorough and constructive feedback, which has significantly improved the clarity of our manuscript. 

  1. Integration of Methodological Relevance by Application: 
    In response to the suggestion to functionally align analytical methods with specific clinical or research applications, we have added a new section titled “Concluding Recommendations and Best-Fit Methodologies”. This section provides a structured comparison of salivary α-amylase (SAA) assessment methods across three key domains: (1) wearable stress monitoring, (2) metabolic phenotyping in obesity, and (3) forensic or retrospective evaluation. We summarize which platforms are recommended, acceptable with caveats, or not recommended, based on current evidence for reproducibility, specificity, and translational value. This integration aims to support informed methodological choices and reduce reliance on poorly validated tools. 
  1. Technical Revisions: 
    A full technical audit of the manuscript has been performed. 
  1. References: Missing references [17] and [41] have been corrected—either restored with full citation or removed where no longer relevant. 
  1. Section Numbering: Subheading numbering, particularly in Chapter 3, has been carefully revised for logical consistency and compliance with formatting standards. 

We trust these revisions address the reviewer’s concerns and enhance the scientific and practical value of the manuscript .

Reviewer 2 Report

Comments and Suggestions for Authors

The review by Gita Erta, Gita Gersone, Antra Jurka and Peteris Tretjakovs is devoted to the analysis literature data on the use of salivary α-amylase as a biomarker of metabolic diseases. Data on analytical methods for determining the concentration of the enzyme in saliva are provided in detail, all existing methods from colorimetric assays to modern technologies be briefly but succinctly listed. Despite the existing limitations in the use of the SAA as a biomarker, which the authors write about, the emergence and development of innovative methods listed in paragraph 4 is observed. This means that attention is paid to the development of this topic and it is promising. Especially considering that taking samples for research occurs in a non-invasive way.

SAA as a biomarker is a rather complex object, since its level is affected by many factors, including genetic, physiological, environmental conditions or psychological state. Clear instructions for sample preparation and recommendations for patients are required from developers of determination methods. Therefore, long-term and thorough research in this area is required, and a large amount of data must be collected to assess the multifactorial impact on this biomarker.

While the manuscript is well presented, it could be improved a little.

In the title, the authors make a statement that the “SAA is a metabolic biomarker in obesity:”, but the review lists many different diseases in which the level of this enzyme may indicate pathology. In my opinion, it would have been possible to limit it to “the enzyme as a metabolic biomarker:” and then everything that is written would have been more consistent with the content.

It should be noted that the list of References is carelessly formatted; references 17 and 41 are missing; it is not clear whether they exist and what the authors are referring to.

Author Response

We thank the reviewer for the thoughtful comments and valuable suggestions. 

  1. Title Revision for Consistency: 
    In response to the reviewer’s recommendation, we have revised the title to better reflect the broader scope of the manuscript. The new title, “Salivary α-Amylase as a Metabolic Biomarker: Analytical Tools, Challenges, and Clinical Perspectives”, aligns more accurately with the diverse pathophysiological contexts discussed in the review while preserving the focus on metabolic relevance. 
  1. Reference Corrections: 
    We have carefully reviewed and corrected the reference list, ensuring consistency in formatting. The previously missing references [17] and [41] have been addressed—either reinserted with complete citation details or removed where no longer applicable.